# Some animals are more equal than others: Validation of a new scale to measure how attitudes to animals depend on species and human purpose of use

**Alexander Bradley**[1,2], **Neil Mennie**[3], **Peter A. Bibby**[1], **Helen J. Cassaday**[1]*

**1** School of Psychology, University of Nottingham, Nottingham, United Kingdom, **2** Department of Education and Sociology, University of Portsmouth, Portsmouth, United Kingdom, **3** School of Psychology, University of Nottingham Malaysia Campus, Kuala Lumpur, Malaysia

* helen.cassaday@nottingham.ac.uk

**Data Availability Statement:** All relevant data are within the paper and its Supporting Information

## Abstract

Globally, many millions of animals are used by humans every year and much of this usage causes public concern. A new scale, devised to measure attitudes to animal use in relation to the purpose of use and species, the Animal Purpose Questionnaire (APQ), was completed by in total 483 participants, 415 British nationals and 68 participants from 39 other countries. The APQ was presented in two survey formats, alongside an established Animal Attitudes Scale (AAS). In both surveys, participants also provided demographic details to provide a context to their attitudes to animals. As might be expected, and consistent with the validity of the new scale, overall scores on the AAS and APQ were highly correlated. However, the APQ provided a more differentiated measure of attitudes to animal use across a variety of settings. The results showed that there was overall higher levels of agreement with the use of animals in medical research and basic science, less endorsement for food production and pest control, and the use of animals for other cultural practices was generally disapproved of, irrespective of species. Participants overall disagreed with the use of rabbits, monkeys, badgers, tree shrews (survey 1), chimpanzees, dogs, dolphins and parrots (survey 2), but were neutral about the use of rats, mice, pigs, octopus, chickens, zebrafish (survey 1), carp, chickens, pigs, pigeons, rabbits and rats (survey 2). Interactions between species and purpose were largely driven by the consideration of using diverse species for food production. In general, females and vegetarians expressed less agreement with the use of animals with some differences by purpose of use. Pet keeping consistently predicted reduced willingness to use animals for basic science (only). The APQ provides a new tool to unpack how public attitudes depend on the intersectionality of demographics, species and purpose of use.

files (with some improved labelling which relates to the captions now provided).

**Funding:** Funding for this study was provided by the School of Psychology, University of Nottingham. The School of Psychology had no further role in study design; in the collection, analysis and interpretation of data; in the writing of the report; and in the decision to submit the paper for publication.

**Competing interests:** The authors have declared that no competing interests exist.

# Introduction

Experimental research involves a global annual usage of over 100 million animals [1]. Estimates of the corresponding figure for farmed animals are even higher (over 50 billion). There is widespread public concern about the use of animals, and a wide-ranging interdisciplinary research agenda [2]. Public attitudes to animal use are an important metric for diverse stakeholders. However, whilst we have some understanding of the role of demographic factors [3–8], there is relatively little systematic evidence as to how public attitudes may depend on species and purpose of use.

Fundamentally, the legislation (in Europe and US) discriminates by species in that (across a variety of purposes of use) some species are more protected than others. For example, animals on the Convention on International Trade in Endangered Species of Wild Fauna and Flora (CITES) list of specially protected species should not be used for any purpose. When considering potential use in research laboratories, non-human primates are afforded higher levels of protection than rodents. Indeed, the US Animal Welfare Act (1970) specified that rats, mice, and birds are not animals under the legislation governing the protection of animals in research. Moreover, although not legally prohibited, with the exception of rabbits, pet species and companion animals are rarely considered to be suitable for human consumption, at least in the UK and US. Pet species such as dogs and cats can have a different status to fish or chickens ('produce'), and rats or mice ('pest'). Wider public opinions on the use of animals are also likely to be influenced by the purpose of use as well as demographic factors. Previous research studying attitudes to animals has focused on specific relationships, such as ownership of an animal as a household pet or for recreational purposes such as horse riding [9] or the use of animals in laboratories [8,10]. However, measuring attitudes with a narrow focus fails to encompass the wider context of human relationships with and attitudes to animals and there is a need for a more systematic approach [11].

## Public concern for animals in relation to purpose of use

To be against the use of animals can manifest in many ways, from opposing animal use for 'frivolous' causes, such as testing cosmetics (which 35% of the population still believe is permitted in the UK [8]), to opposing any use of any animal whatsoever. The use of animals in biomedical research is highly regulated. The UK Home Office reports annually on the profile of animal usage for research purposes [12]. In 2018, the majority of procedures involved mice, fish and rats, which together amounted of 93% of the experimental procedures conducted.

Despite the potential benefit to humans, biomedical research raises significant public concern. A study of 21 books written by animal rights activists analysed the number of pages showing concern for different uses of animals: 63.3% of pages were about the use of animals in research and education, whilst these activities account for approximately 0.3% of animal use in total [13]. However, this profile of concern may not be representative of that shown by other publics. For example, postal questionnaire methods to reach a random sample of the population of Northern Ireland, suggested that the highest levels of concern were shown in relation to activities expected to lead to the animal's death or injury (hunting and fighting in addition to research) as compared with activities resulting in less suffering [14].

## Public concern in relation to species in use

Attitudes to animal use can also vary widely depending on the perceived characteristics of the species in question which in turn influence their emotional responses and perceptions of the animal's instrumental value to humans [15]. Discrimination based on species membership or 'speciesism' [16,17] of one kind or another is widespread: conservation efforts are often greater

or at least more widely publicised for large, attractive mammals, particularly those who share some similarity with humans, for example other primates [18]. Perceived cognitive ability is also a likely moderating variable and human estimates of the intelligence of other species generally correspond with the ordering of the phylogenetic scale [19]. Species perceived to be cognitively able such as primates and pets are viewed with higher ethical regard, whilst the use of species perceived as cognitively less able, such as fish, is reported to be more acceptable [7]. Mere reputation of a species' behavioural attributes can also influence attitudes even if founded on prejudice [3,20]. If a species is perceived as dangerous or vicious, humans tend to perceive it more negatively. Human reactions have even been found to vary in relation to more arbitrary factors such as body size, as well as cultural and historical relationships, familiarity, and utility value amongst other factors [21,22]. Relatedly, the utilitarian or affectionate labels given to animals have been reported to influence perceptions of their moral standing [23]. Thus, judgements of the sentiency of different species of animal and hence the legitimacy of their use for human purposes may be based on quite limited information and subject to cultural differences [19,24].

## Moderating effects of diet and gender

Arguably, individuals most against the exploitation of animals are those that incorporate this attitude into their lifestyles, for example vegetarians and vegans [25,26]. Vegetarianism and other dietary choices have been linked to gender: female outnumber male vegetarians in Western society and those females who do eat meat eat relatively less [27]. Ipsos MORI is a UK-based market research company which conducts independent surveys for a wide range of organisations. In line with the gender difference in attitudes to eating meat, the latest Ipsos MORI poll showed that 30% of males in the UK are unconcerned about the use of animals in experiments compared with 14% of women [8]. Questionnaire measures of attitudes to animal use also show that being female is associated wth higher levels of expressed concern for animals [4–6]. Regression analyses suggest that gender accounts for 19.5% of the variance in attitudes towards animals [28].

## The need for additional measures of attitudes to animal use

The Animal Attitudes Scale (AAS) [4] has been widely used to examine attitudes to animal use in different contexts and is now available in a shortened form which has good reliability and validity (AAS-10) [29]. The AAS-10 names a number of species (cattle and hogs, whales and dolphins, cats, dogs and rabbits) and potential uses of animals (e.g. medical research, human consumption, clothing, education, safety testing, pet keeping and zoos) but there is no systematic comparison of attitudes by purpose of use.

Qualitative data has underscored the importance of a multi-dimensional approach to understanding attitudes to animal use. For example, thematic analysis of interview data identified the importance of 'types of animal used', 'purpose of animal use' and 'knowledge of animal use' were important moderators of attitudes to animal use [30]. Using a quantitative approach, attitudes to different purposes of use (medical research, dissection, personal decoration and entertainment) have been examined in different populations (scientists, animal welfarists and laypersons) using 4 questionnaire variants to measure attitudes to (1) monkeys, (2) dogs and cats, (3) rabbits and guinea pigs or (4) rats and mice [31]. However, in this earlier study each participant was asked about their attitudes to the use of only one of the 4 sets of species. Likewise, categories of purpose have been examined with a Pet, Pest and Profit (PPP) scale [32]. Similar to the APQ, the PPP provides a multi-dimensional alternative to the AAS but a variety of purposes are subsumed under profit. Thus, although the available instruments may capture

comparative data on the use of animals for different purposes and although some differences by species and purpose have been examined, the intersectionality between species and purpose of use has not.

## The present study

To address this gap, the present article reports two studies using a newly devised Animal Purpose Questionnaire (APQ) to systematically compare attitudes to animal use across different purposes of use. We also compared attitudes to the (hypothetical) use of a diverse range of species, including exemplars of those falling into the categories of pet, pest and profit [32,33] as well as some non-domesticated species. The specific animal species included were mice, rats, rabbits, pigs, monkeys, octopus, chickens, badgers, zebrafish and tree shrews (survey 1) and carp, chicken, chimpanzee, dog, dolphin, frog, parrot, pig, pigeon, rabbit, rat and snake (survey 2). Survey 1 was a pilot study and survey 2 was a follow-up validation study.

Zebrafish and tree shrew were initially included to examine attitudes in relation to species that are less known to the general public. Zebrafish have been very important for biological research and used relatively often, but publicity of their findings in main stream media are minimal. The tree shrew was included because it is an unfamiliar animal to most and whilst the name suggests a rodent-like creature, the tree shrew is a small mammal with a relatively large brain, closely related to the primate species. Survey 2 used a revised selection of species which would be more readily identified by participants.

Demographic factors such as educational level, age, early experience with pets, ethnicity and religious affiliation have also been found to be important determinants of attitudes to animal use [7,8,34] and gender is a better predictor than personality [5,28]. We therefore also included a few questions on basic demographics, including gender and diet.

Thus, although the primary objective was to validate the new APQ scale, the data collected in the present studies were also used to examine a number of hypotheses. Firstly, participants who self-report strict non-meat-eating dietary preferences would be expected to show higher disagreement with animal use. Secondly, amongst the purposes examined by the APQ, it was to be expected that, for all species, participants would show the highest agreement with the use of animals in medical research because of the potential benefits to the quality and duration of human lives [8,34,35]. Finally with reference to speciesism [16,17], participants were predicted to be relatively more accepting of the use of animal species which are seen to be less intelligent, less emotional and less cognitively capable, especially in medical research [30,31]. For example, participants would be expected to show the lowest acceptance of use of animals with high intelligence (e.g., dolphins), those which relate to humans socially as pets (e.g., dogs) and those which are genetically similar to humans (e.g., chimpanzees).

## Methods

Both studies were approved by the University of Nottingham UK School of Psychology Ethics Committee (Ref: 994R and Ref: S1021). The circulation of survey 2 was also approved by the University of Nottingham Malaysia Ethics Committee (application number NRM08122017). There were some generic examples of categories of purpose but there were no examples of how specific animals may be used in practice (and participants were advised that some permutations of species and use were hypothetical).

## Participants

Undergraduate participants were recruited using the University of Nottingham UK School of Psychology Research Participation Scheme (Sona Systems® software; $N$ = 65 for survey 1) and

by convenience sampling ($N$ = 63 for survey 1). To reach a wider population, the survey 2 link was distributed through social media and further propagated by snowballing to improve the diversity of the sample (the link was accompanied by a brief description of the study and the encouragement to share with others who might find it interesting). The survey 2 link was also posted to vegan and vegetarian forums (UK Vegan, Nottingham Vegans, Vegetarian Society), to increase the proportion of non-meat-eaters included in the sample. It was further disseminated via the University of Nottingham Malaysia Campus, to target an Eastern population and improve the cultural diversity of participants sampled.

In survey 1, a total of 128 participants (M = 39, F = 89) agreed to participate. The modal age group of the participants was '21 and under' ($N$ = 88). The remainder of the sample were aged '22 to 34' ($N$ = 32), '35 to 44' ($N$ = 3), '45 to 54' ($N$ = 2), '55 to 64' ($N$ = 1), '65 or over' ($N$ = 2). The nationality of participants was 115 British and 13 other nationalities. In survey 2, 419 participants finished the on-line survey and provided some demographic details. Participants had a mean age of 33.84 (SD = 15.71) and were predominantly female ($N$ = 314, 75%). However, of these 419 participants, around 64 did not complete all parts of the APQ ($N$ = 355 for the mixed ANOVA). Similar to survey 1, the majority of participants were British ($N$ = 300) but the proportion of British nationality was lower (71% compared with 89.8% in survey 1) and the rest of participants were of a wider range of (39) other nationalities. In both surveys, participants were presented with four options in relation to gender: male, female, 'other' and 'prefer not to say'. In survey 1, all participants self-identified themselves as male or female. In survey 2, one selected 'other' and two participants selected 'prefer not to say' (these 3 participants were excluded for analyses by gender). Further demographic details are shown in Table 1 (as percentages for comparison across the two surveys).

## Materials

An information sheet was provided, in both survey formats, to provide a brief description of the questionnaires to follow. All participants gave informed consent before starting the surveys. After completion of the questionnaires (see below), a separate page or screen requested that participants provide some demographic details: gender, age, nationality, dietary preferences, pet ownership and level of education. Where fixed response options were provided,

**Table 1. Participant demographics for surveys one and two.**

|  | Survey 1 | Survey 2 |
|---|---|---|
| **Gender (% Female)** | 70% | 75% |
| **Age** | Mode = 18–21 | Mean = 33.84 |
|  | Range = 18–65+ | SD = 15.71 |
| **Nationality (% British)** | 89.8% | 71% |
| **Diet** |  |  |
| Meat Eaters (%) | 74% | 55% |
| Vegetarians (%) | 14% | 9% |
| Vegan (%) | 6% | 7% |
| Pescatarians (%) | 4% | 4% |
| Other (%) | 2% | 25% |
| **Pet Ownership (%)** | 81% | 88% |
| **Received Education Level** |  |  |
| School level: GCSE A level or equivalent | 61% | 73% |
| Degree or Equivalent | 17% | 9% |
| Postgraduate Study (%) | 22% | 6% |

these included 'other' and/or 'prefer not to say', as appropriate. Participants were also given the opportunity to provide any open-ended comments or questions they would like noted at this point. Finally, they were provided with a written debrief which described the objectives of the study in more depth. The survey 2 data was collected through Qualtrics, an online platform for designing and distributing surveys (1st December 2017 to 17th August 2018).

**Animal attitude scale.** The AAS-10 is the shorter 10 item version of the original 20-item AAS [4]. Each item is rated on a five-point scale (1 = strongly disagree, 2 = disagree, 3 = undecided, 4 = agree, and 5 = strongly agree). Five of the 10 items were reverse scored; higher total scores reflect higher levels of concern for animals. The reported reliability of the short form of the scale is high, Cronbach's alpha = 0.90 [29]. A high score indicates a pro-welfare attitude towards animals.

**Animal purpose questionnaire.** In survey 1, the APQ items were headed by examples to illustrate the designated categories of purpose: medical research (e.g., for an animal model of dementia); basic science (e.g., to better understand the brain); food production (e.g., 'bush meat' in the tropics where applicable); pest control (e.g., when crops are damaged); other cultural practices (e.g., the use of body parts as ornaments). There was also the instruction to consider any other uses which might seem to fall within each of the categories (the survey 1 format is shown in S1 Appendix).

Participants were required to rate their level of agreement with the use of different types of animal for the different categories of purpose by checking the appropriate box on 5 point rating scales (1 = strongly disagree, 2 = disagree, 3 = undecided, 4 = agree, and 5 = strongly agree). Thus their responses were used to indicate levels of agreement or disagreement with any kind of use (within each of the five broad categories) which directly or indirectly results in the killing of each of the species examined. A high score reflects agreement with use of animals, thus for the APQ a low raw score indicates a pro-welfare attitude towards animals.

For survey 1, the experimenters used set definitions [36] if participants had any queries about the species presented (e.g. tree shrew and zebrafish). The species included in the online version of the APQ were pig, chicken, dog, dolphin, chimpanzee, rabbit, rat, snake, frog, pigeon, carp (fish) and parrot. This was a slightly different selection of species to that used in the paper-based survey 1. The layout was also improved for the online version to encourage comparative judgements: the five purposes were presented one at a time and participants were asked to provide their responses for each of the 12 animals sampled (the survey 2 format is shown in S2 Appendix). The order of animals presented to each participant was randomised.

The species included for the online phase of the study were selected as broadly plausible for the range of purposes examined. The online phase of the study also included a selected fish species–carp, the mostly widely recognised farmed fish [37]. The final selection of species was also judged to be suitable for use in other cultures. The purposes included were the same as those in the survey 1 APQ, with some improved clarification in survey 2: food production, e.g. any form of commercial or domestic consumption of animal product; pest control (the removal of an animal that is impacting adversely on human activity), e.g. to reduce damage to crops or homes; other uses, e.g. for hunting or fighting, for sport, for wearing (the skin) as clothing or as ornamentation, for display as a trophy. In both surveys, the APQ items were prefaced with the instruction to '. . .*rate whether you agree or disagree with the killing of different types of animal for the following purposes. . .*'

**Bem sex role inventory.** The Bem Sex-Role Inventory (BSRI) scale provides a continuous measure of psychological profiles pertaining to traits traditionally associated with masculinity and femininity [38,39]. We used a 12 item short form of the BSRI [40,41]. Each item was rated on a 7 point scale from 1 (never or almost never true) to 7 (almost always true). Six of the items measured feminine traits and 6 measured masculine traits. The BSRI was not administered in survey 2.

## Procedure

Both surveys began by asking participants to read an information sheet and then—if willing to proceed–to complete a consent form. Participants then completed the questionnaires, these were paper-based in survey 1 and presented online via Qualtrics in survey 2. Both surveys requested that participants provide the demographic information at the end. For the online survey, participants sent their responses by pressing a submit button. Upon completion of the study participants were thanked and informed of the purpose of the research. A debrief provided clarification of the experimental objectives, the planned, further reading and contact links for further information.

## Design and analysis

The study used a repeated measures design in that all participants complete the same 3 questionnaires and were asked to provide the same additional information under the same conditions. For the APQ, repeated measures Analysis of Variance (ANOVA) was used to examine attitudes by species separately by the use. Negatively worded items in the AAS were reverse coded prior to analyses in SPSS and R.

There was no reverse scoring of the APQ scale. Those with a high regard for animals will receive a low APQ score as their high concern for animals leads them to strongly disagree with the use of animals regardless of the purpose of that use. In the case of the AAS, participants who have a high concern for animals will score highly because they strongly agree with statements such as 'I sometimes get upset when I see wild animals in cages at zoos' (and negatively worded items were reverse coded). There should therefore be a negative correlation between scores on the AAS and APQ scales.

For the APQ in survey 1 (only), a 'not applicable' choice option was provided and scored 2.99 (equivalent to the median rating of neutral) in order that the participants' other responses were not excluded (because of the repeated measures nature of the design used to test for interactions between species and purpose of use). This response option was dropped for the survey 2. Unless otherwise stated the comparisons between means and the correlations reported in this paper are corrected using Benjamini and Hochberg's procedure [42] for controlling the false discovery rate rather than the Bonferroni method (which can be overly conservative when there are large numbers of tests leading to less statistical power). All post hoc comparisons reported as significant or non-significant in the text were made at the $p \leq 0.05$ type 1 error rate.

Significant correlations between the APQ, BSRI measures and demographic variables were followed up with regression analyses. Convergent validity between the APQ and the AAS was tested for using Pearson correlations.

## Results

### Survey 1 pilot study

All the participants' ($N$ = 128) data was used in the analyses (the data are provided as S1 Data).

Differences by gender, species and purpose of use as measured by the APQ

The test for sphericity was statistically significant for the species and purpose effects. Since the epsilon was less than .75 Greenhouse-Geisser corrected values are reported. Corrected degrees of freedom are shown to the nearest integer.

A three-way (2x10x5) mixed analysis of variance was conducted with gender, species and purpose as the independent variables. The dependent variable was participants' responses to the APQ items. The means and standard errors are shown in Table 2.

There was a main effect of gender ($F_{1,126}$ MSe = 28.559, p<0.001, $\eta_p^2$ = 0.209). Overall, males were marginally inclined to agree to the use of animals (M = 3.361, s.e.m. = .080) while the females were inclined to disagree (M = 2.523, s.e.m. = .121). There was also main effect of species, ($F_{6,773}$ = 40.472, MSe = 1.648, p<0.001, $\eta_p^2$ = 0.243) and a main effect of purpose ($F_{3,324}$ = 130.988, MSe = 5.656, p = < .001, $\eta_p^2$ = 0.51). The means (s.e.m.) are as follows: mice = 3.276 (.079), rats = 3.291 (.084), rabbits = 2.897 (.084), pigs = 3.044 (.080), monkeys = 2.321 (.080), octopus = 2.996 (.085), chickens = 3.027 (.082), badgers = 2.668 (.083), zebrafish = 3.088 (.096), tree shrews = 2.815 (.089). Overall, participants tended to disagree with the use of rabbits, monkeys, badgers and tree shrews but neither agree nor disagree with the use of rats, mice, pigs, octopus, chickens or zebrafish. The means (s.e.m.) are as follows: medical research = 3.630 (.089), basic science = 3.542 (.092), food production = 2.796 (.086), pest control = 2.727 (.091) and other cultural practices = 2.107 (.087). Participants overall tended to agree with using animals for medical and basic science research, disagree with using animals for other cultural practices and neither agree nor disagree with using animals for food production and pest control.

There were two statistically significant two-way interactions: gender by animal ($F_{6,773}$ = 3.721, MSe = 1.648, p = 0.001, $\eta_p^2$ = 0.029; see Fig 1) and animal by purpose ($F_{14,1821}$ = 29.245, MSe = 0.86, p < .001, $\eta_p^2$ = 0.188; see Fig 2). For the gender by animal interaction the differences between male and female participants were statistically significant for all the animals. The largest difference (1.107) was for rabbits and the smallest for monkeys (0.538).

The picture for the animal by purpose interaction is somewhat more complicated. However, some patterns are relatively clear. A consistent pattern was found in relation to other cultural practices, such that for all animals participants disagreed with such other uses. For pigs and chickens it was generally agreed that it is acceptable to use these animals for food production. For the majority of the other animals this was not the case; participants disagreed with food production use. The exceptions were octopus and zebrafish. For these animals participants neither agreed nor disagreed with their use in food production. For all animals, except monkeys and badgers, participants tended to agree with the use of animals for medical and basic science research. For monkeys and badgers, participants neither agreed nor disagreed with using these animals for medical and basic science research. For pest control, participants were against or indifferent to the animals' use, with the exception of rats and mice. In the latter case participants tended to agree.

The three-way interaction between gender, animal and purpose was also significant ($F_{14,1821}$ = 5.594, MSe = 0.86, p < .001, $\eta_p^2$ = 0.043). This suggests that overall the complex relationship between animal and purpose are not always the same for males and females. However, as can be seen in Table 2, such differences tended to be quite small and as the overall patterns were similar across gender, they will not be discussed further.

Correlational and regression analyses to explore the interrelationships between questionnaire scores and demographic factors

To reduce the complexity of the analysis, the responses to the 10 animals used for the species questions were factor analysed for each of the five different purposes separately. In each case the solution was dominated by a single factor (min % variance = 59.9, max % variance = 75.3%). Thus, the mean responses of the species for the different purposes were calculated and their reliabilities assessed: medical research α = .96, basic science α = .96, food production α = .93, pest control α = .94 and other cultural practices α = .96. Reliabilities were also calculated for each of the animal types across the five purposes. All the Cronbach's alphas were between .81-.89 indicating good internal consistency between items for each animal type. The overall scale also showed good internal consistency (α = .98).

**Table 2. The mean (and standard errors) of the purpose items of the APQ by participants' gender, purpose of use and species of animal, for survey 1 and survey 2.**

| Survey 1 | Female | | | | | Male | | | | |
|---|---|---|---|---|---|---|---|---|---|---|
| | MR | BS | FP | PT | OCP | MR | BS | FP | PT | OCP |
| Mice | 3.663 (0.11) | 3.416 (0.114) | 2.235 (0.124) | 2.853 (0.118) | 1.752 (0.121) | 4.487 (0.166) | 4.385 (0.173) | 3.051 (0.187) | 4.179 (0.179) | 2.743 (0.183) |
| Rat | 3.663 (0.109) | 3.416 (0.113) | 2.258 (0.133) | 2.842 (0.127) | 1.831 (0.125) | 4.538 (0.165) | 4.385 (0.17) | 3.076 (0.201) | 4.128 (0.191) | 2.769 (0.189) |
| Rabbit | 2.91 (0.122) | 2.831 (0.119) | 2.112 (0.128) | 2.213 (0.128) | 1.652 (0.107) | 4.231 (0.184) | 4.051 (0.18) | 3.41 (0.194) | 3.384 (0.194) | 2.179 (0.161) |
| Pig | 3.124 (0.116) | 2.955 (0.119) | 3.112 (0.135) | 2.224 (0.122) | 1.719 (0.108) | 4.231 (0.175) | 4.051 (0.18) | 4.077 (0.204) | 2.768 (0.184) | 2.179 (0.164) |
| Monkey | 2.596 (0.132) | 2.629 (0.143) | 1.595 (0.102) | 1.899 (0.113) | 1.539 (0.087) | 3.795 (0.2) | 3.513 (0.216) | 2.051 (0.154) | 1.923 (0.17) | 1.667 (0.131) |
| Octopus | 3.157 (0.112) | 3.27 (0.113) | 2.596 (0.132) | 2.303 (0.132) | 1.708 (0.115) | 4.128 (0.169) | 4.077 (0.171) | 3.795 (0.2) | 2.563 (0.2) | 2.359 (0.173) |
| Chicken | 3 (0.113) | 3.011 (0.114) | 3.315 (0.136) | 2.213 (0.13) | 1.708 (0.113) | 4.128 (0.171) | 4.128 (0.172) | 4.077 (0.205) | 2.538 (0.196) | 2.154 (0.171) |
| Badger | 2.697 (0.124) | 2.629 (0.128) | 1.921 (0.115) | 2.292 (0.129) | 1.629 (0.1) | 3.949 (0.187) | 3.821 (0.194) | 2.435 (0.174) | 3.231 (0.195) | 2.076 (0.151) |
| Zebrafish | 3.157 (0.13) | 3.236 (0.125) | 2.506 (0.139) | 2.371 (0.131) | 1.843 (0.126) | 4.102 (0.196) | 4.051 (0.189) | 3.512 (0.21) | 3.307 (0.198) | 2.794 (0.19) |
| Tree Shrew | 3.067 (0.118) | 3.079 (0.12) | 2.18 (0.123) | 2.404 (0.138) | 1.831 (0.117) | 3.974 (0.178) | 3.897 (0.182) | 2.615 (0.186) | 2.897 (0.209) | 2.205 (0.176) |
| Survey 2 | Female | | | | | Male | | | | |
| | MR | BS | FP | PT | OCP | MR | BS | FP | PT | OCP |
| Carp | 2.57 (0.07) | 2.45 (0.07) | 2.53 (0.08) | 1.96 (0.06) | 1.87 (0.06) | 3.16 (0.15) | 3.08 (0.14) | 3.07 (0.14) | 2.43 (0.12) | 2.45 (0.13) |
| Chicken | 2.55 (0.07) | 2.46 (0.07) | 2.91 (0.08) | 1.77 (0.06) | 1.83 (0.06) | 3.15 (0.14) | 3.05 (0.13) | 3.81 (0.14) | 2.12 (0.12) | 2.46 (0.13) |
| Chimpanzee | 2.17 (0.07) | 2.11 (0.07) | 1.3 (0.04) | 1.48 (0.05) | 1.50 (0.04) | 2.59 (0.15) | 2.50 (0.14) | 1.42 (0.07) | 1.69 (0.09) | 1.87 (0.11) |
| Dog | 2.11 (0.07) | 2.04 (0.06) | 1.39 (0.04) | 1.67 (0.05) | 1.54 (0.04) | 2.57 (0.15) | 2.48 (0.14) | 1.72 (0.10) | 1.96 (0.11) | 1.96 (0.11) |
| Dolphin | 1.96 (0.06) | 1.96 (0.06) | 1.39 (0.04) | 1.46 (0.05) | 1.46 (0.04) | 2.23 (0.14) | 2.27 (0.14) | 1.50 (0.08) | 1.66 (0.09) | 1.68 (0.09) |
| Frog | 2.54 (0.07) | 2.46 (0.07) | 1.88 (0.07) | 2.15 (0.07) | 1.84 (0.06) | 3.17 (0.14) | 3.15 (0.14) | 2.33 (0.13) | 2.50 (0.13) | 2.49 (0.13) |
| Parrot | 2.28 (0.07) | 2.17 (0.07) | 1.51 (0.05) | 1.70 (0.05) | 1.63 (0.06) | 2.70 (0.14) | 2.67 (0.13) | 1.91 (0.12) | 2.08 (0.11) | 2.10 (0.12) |
| Pig | 2.55 (0.07) | 2.44 (0.07) | 2.80 (0.08) | 1.80 (0.06) | 1.80 (0.05) | 3.19 (0.15) | 3.06 (0.14) | 3.68 (0.14) | 2.09 (0.12) | 2.38 (0.13) |
| Pigeon | 2.60 (0.07) | 2.49 (0.07) | 2.12 (0.07) | 2.60 (0.08) | 1.86 (0.06) | 3.14 (0.14) | 3.15 (0.14) | 2.62 (0.14) | 2.95 (0.14) | 2.49 (0.14) |
| Rabbit | 2.48 (0.07) | 2.36 (0.07) | 2.23 (0.07) | 1.96 (0.06) | 1.74 (0.06) | 3.16 (0.14) | 3.08 (0.14) | 3.01 (0.14) | 2.57 (0.13) | 2.36 (0.13) |
| Rat | 2.82 (0.08) | 2.66 (0.08) | 1.73 (0.06) | 3.17 (0.08) | 1.97 (0.07) | 3.52 (0.14) | 3.41 (0.14) | 2.18 (0.13) | 3.55 (0.14) | 2.70 (0.14) |
| Snake | 2.50 (0.07) | 2.37 (0.08) | 1.70 (0.06) | 2.58 (0.08) | 1.83 (0.06) | 3.25 (0.14) | 3.10 (0.14) | 2.06 (0.12) | 2.89 (0.13) | 2.47 (0.13) |

*Note*: MR = Medical Research, BS = Basic Science, FP = Food Production, PT = Pest Control, OCP = Other Cultural Practices.

The correlations between the questionnaire score (AAS and APQ) and the relevant demographic variables are shown in Table 3. Age, level of education and nationality of the participants are not considered as these variables were not correlated with the AAS and AAQ

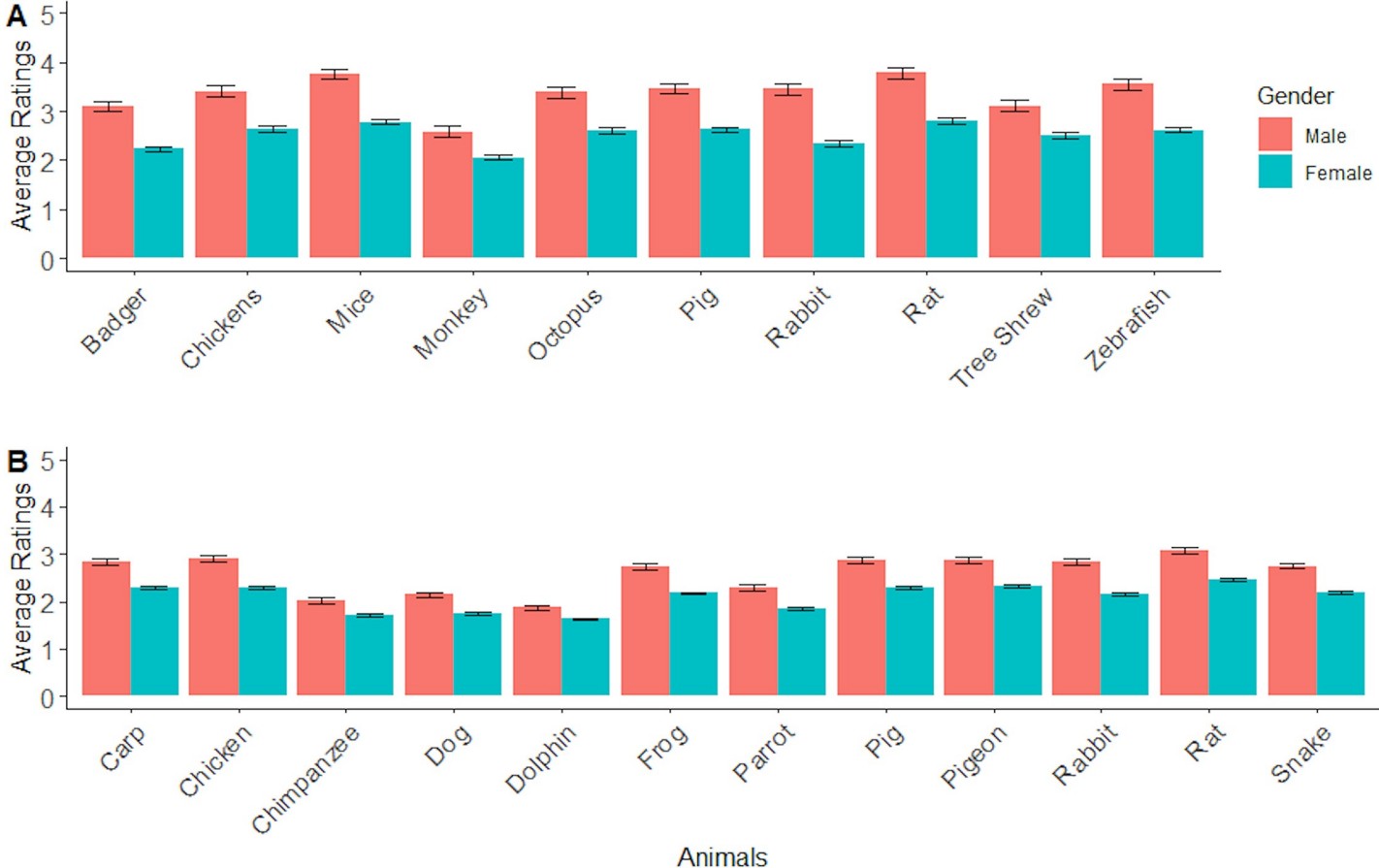

**Fig 1.** Mean levels of (dis-)agreement with the use of animals, as measured by the Animal Purpose Questionnaire (APQ) ratings, shown for male and female participants for each of the different animal species, for (A) survey 1 and (B) survey 2. Error bars show the standard errors of the means.

measures of interest. Gender was not correlated with diet; there was no evidence that males and females differed in their tendency to eat meat. There was a significant negative relationship with having a pet, with females significantly more likely to report having a pet. Gender was significantly positively correlated with BSRI masculinity and negatively correlated with BSRI femininity indicating that males scored higher on masculinity and lower on femininity than females. The negative correlation between gender and AAS showed that males score significantly lower than the females on the AAS. For each of the APQ purpose scales there was a significant positive correlation between gender and the scale ratings. The previous analyses further explicated these relationships.

Diet, that involved animals or not, was negatively correlated with the AAS score and positively correlated with the APQ scores, with the exception of other cultural practices. Participants who reported eating meat have a less favourable attitude to animals and are more willing to endorse their use for medical and basic science research, food production and pest control. A similar, but opposite pattern, was found for pet ownership. Participants reporting pet ownership had a more favourable attitude toward animals and were less likely to endorse their use for medical and basic science research, pest control and other cultural practices.

Apart from the correlation with gender, the masculinity component of the BSRI was only significantly correlated with the AAS score. This positive relationship suggests that those who scored higher on the masculinity scale have a more favourable attitude to animals however,

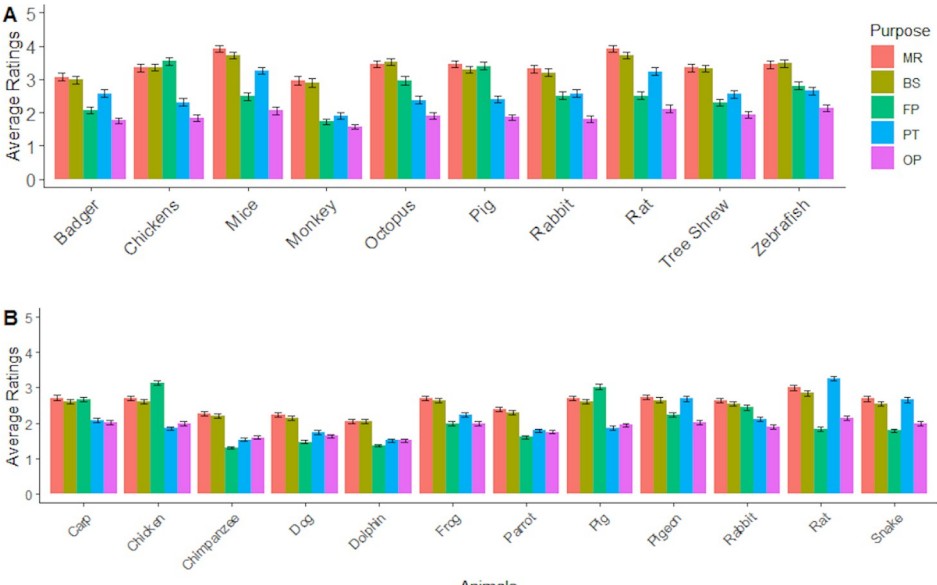

**Fig 2.** Mean levels of (dis-)agreement with the use of animals, as measured by the Animal Purpose Questionnaire (APQ) ratings, shown by suggested purposes of use for each of the different animal species, for (A) survey 1 and (B) survey 2. MR = Medical Research, BS = Basic Science, FP = Food Production, PT = Pest Control, OCP = Other Cultural Practices. Error bars represent the standard error of the means.

masculinity did not correlate with any of the APQ purpose of use scales. The femininity component was positively correlated with the AAS score and negatively correlated with all the APQ purpose scales. In other words, participants scoring higher on femininity show more favourable attitudes to animals and are also less likely to endorse using animals for medical and basic science research, food production, pest control and other cultural purposes.

All the APQ purpose scales were significantly negatively correlated with the AAS score (min $r$ = -.44, max $r$ = -.63) As might be expected, less favourable attitudes to animals was associated with a greater willingness to use animals. All the APQ purpose scales were also positively correlated with each other but varied in the strength of the correlation (min $r$ = .39, max $r$ = .94). Overall, the more participants felt inclined to let animals to be used for one of the five purposes measured herein, the more inclined they were to let animals be used for another purpose.

To further explore how the demographic and the BSRI measure relate to the purpose of use scales five multiple linear regression were conducted with the APQ purpose scales for medical and basic science research, food production, pest control and other cultural practices as the criterion variables. In each analysis, gender, diet, pet ownership, BRSI scores were simultaneously entered into the regression model. The results are shown in Table 4.

All five models had significant F-tests ($p$ < .001) suggesting that the models were a better fit than the empty models (model with just the intercept included). Furthermore, predictors in the model explained between 16–37% of the variability in APQ purpose scores with the most amount of total variability explained in food production ($R^2$ = .37), medical research ($R^2$ = .29) and basic science ($R^2$ = .28). As expected (see earlier analyses) gender was an important positive predictor for medical science, basic science, food production and pest control suggesting that males were more willing to use animals for these purposes. Diet was a positive predictor for all APQ purpose of use scales except for other cultural practices which suggests that those who eat meat are more comfortable using animals for medical research, basic science, food

**Table 3. The correlations between the questionnaire score (AAS and APQ) and demographic variables, for survey 1 and survey 2.**

| Survey 1 | Gend | Diet | Pet | Masculinity | Femininity | AAS | MR | BS | FP | PT | |
|---|---|---|---|---|---|---|---|---|---|---|---|
| Diet | .01 | | | | | | | | | | |
| Pet | -.33 | .08 | | | | | | | | | |
| Masculinity | .20 | -.10 | .05 | | | | | | | | |
| Femininity | -.39 | -.01 | .17 | .05 | | | | | | | |
| AAS | -.43 | -.36 | .26 | .21 | .21 | | | | | | |
| MR | .46 | .20 | -.30 | -.02 | -.21 | -.63 | | | | | |
| BS | .43 | .21 | -.32 | -.02 | -.25 | -.63 | .94 | | | | |
| FP | .39 | .41 | -.16 | -.09 | -.22 | -.61 | .67 | .67 | | | |
| PT | .34 | .23 | -.20 | -.03 | -.19 | -.52 | .66 | .65 | .78 | | |
| OCP | .29 | .16 | -.25 | -.04 | -.24 | -.44 | .42 | .39 | .60 | .54 | |
| **Survey 2** | Gend | Diet | Pet | Religion | Ethnicity | Age | AAS | MR | BS | FP | PT |
| Diet | .20 | | | | | | | | | | |
| Pet | -.04 | .04 | | | | | | | | | |
| Religion | -.08 | -.00 | -.09 | | | | | | | | |
| Ethnicity | .01 | -.12 | -.21 | .30 | | | | | | | |
| Age | .06 | .06 | .13 | .17 | -.32 | | | | | | |
| AAS | -.26 | -.21 | .08 | -.22 | -.10 | -.13 | | | | | |
| MR | .20 | .14 | -.11 | .12 | .14 | .04 | -.75 | | | | |
| BS | .22 | .13 | -.13 | .14 | .23 | -.03 | -.75 | .86 | | | |
| FP | .24 | .23 | .01 | .09 | -.04 | .14 | -.70 | .62 | .60 | | |
| PT | .17 | .15 | -.08 | .13 | .04 | .14 | -.63 | .64 | .62 | .66 | |
| OCP | .24 | .19 | -.04 | .13 | .13 | .10 | -.68 | .60 | .64 | .60 | .55 |

*Note*: Correlations with a shaded background are statistically significant at p ≤ .05. Gend = Gender (female = 0, male = 1); Diet (does not eat meat = 0, eats meat = 1); Pet (does not have a pet = 0, has a pet = 1); Religion (No Religion = 0, Religion = 1); Ethnicity (White = 0, Non-white = 1); AAS = Animal Attitudes Scale; MR = Medical Research, BS = Basic Science, FP = Food Production, PT = Pest Control, OCP = Other Cultural Practices.

production and pest control purposes than those who do not eat meat. Pet ownership was a significant negative predictor of medical research and basic science indicating that those who own pets were less willing than non-pet owners to use animals for these purposes. Neither BSRI femininity nor BSRI masculinity scores are predictors of the five APQ purpose of use scales once gender is statistically controlled for.

## Survey 2 validation study

Four hundred and nineteen participants completed the questionnaire including the demographic questions. However, 64 of the 419 did not complete all items on the APQ so the mixed effect ANOVA is conducted on 355 participants (the available data are provided as S2 Data).

**Differences by gender, species and purpose of use as measured by the APQ.** The test for sphericity was significant for the effect of purpose and the purpose by gender interaction with an epsilon greater than .75 so Huhn-Feldt corrected values are reported.

A three-way (2 x 12 x 5) mixed ANOVA was conducted with gender, species and purpose as independent variables and purpose of use as the dependent variable. The means and standard errors are shown in Table 2.

There was a main effect of gender ($F_{1, 353} = 24.42$, MSe = 41.10, $p < 0.001$, $\eta_p^2 = .06$) with males being more likely to be neutral to the use of animals (M = 2.60, s.e.m. = 0.02) while females were more inclined to disagree (M = 2.10, s.e.m. = 0.01). There was also a main effect

**Table 4. Results of the multiple linear regression, for survey 1 and survey 2.**

| Survey 1 | MR | | BS | | FP | | PT | | OCP | |
|---|---|---|---|---|---|---|---|---|---|---|
| | β | SE | β | SE | β | SE | β | SE | β | SE |
| Intercept | 3.57*** | 0.57 | 3.79*** | 0.58 | 2.59*** | 0.50 | 2.77*** | 0.58 | 2.65*** | 0.56 |
| Gender | 0.94*** | 0.20 | 0.78*** | 0.21 | 0.78*** | 0.18 | 0.62** | 0.21 | 0.39 | 0.20 |
| Diet | 0.49** | 0.18 | 0.53** | 0.19 | 0.96*** | 0.16 | 0.52** | 0.19 | 0.35 | 0.18 |
| Pet | -0.46* | 0.22 | -0.57* | 0.22 | -0.15 | 0.19 | -0.28 | 0.22 | -0.40 | 0.21 |
| Masculinity | -0.07 | 0.08 | -0.05 | 0.08 | -0.11 | 0.07 | -0.06 | 0.08 | -0.05 | 0.08 |
| Femininity | -0.02 | 0.08 | -0.07 | 0.07 | -0.05 | 0.07 | -0.05 | 0.08 | -0.11 | 0.07 |
| F value | 10.01*** | | 9.47*** | | 14.39*** | | 5.49*** | | 4.55*** | |
| $R^2$ | 0.29 | | 0.28 | | 0.37 | | 0.18 | | 0.16 | |
| Survey 2 | MR | | BS | | FP | | PT | | OCP | |
| | β | SE | β | SE | β | SE | β | SE | β | SE |
| Intercept | 2.14*** | 0.24 | 2.05*** | 0.23 | 1.38*** | 0.18 | 1.65*** | 0.18 | 1.10*** | 0.19 |
| Gender | 0.49*** | 0.14 | 0.52*** | 0.13 | 0.44*** | 0.10 | 0.29** | 0.10 | 0.47*** | 0.11 |
| Diet | 0.34* | 0.15 | 0.33* | 0.14 | 0.40*** | 0.11 | 0.29** | 0.11 | 0.39*** | 0.12 |
| Pet | -0.32 | 0.18 | -0.27* | 0.17 | 0.01 | 0.13 | -0.23 | 0.13 | -0.06 | 0.14 |
| Religion | 0.16 | 0.12 | 0.19 | 0.12 | 0.17 | 0.09 | 0.14 | 0.09 | 0.13 | 0.10 |
| Ethnicity | 0.37* | 0.16 | 0.58*** | 0.15 | -0.03 | 012 | 0.11 | 0.11 | 0.37** | 0.13 |
| Age | 0.01 | 0.00 | 0.01* | 0.00 | 0.01 | 0.00 | 0.01* | 0.00 | 0.01* | 0.00 |
| F value | 6.51*** | | 9.56*** | | 8.66*** | | 6.19*** | | 9.49*** | |
| $R^2$ | .08 | | .12 | | 0.11 | | .08 | | .12 | |

*Note*: MR = Medical Research, BS = Basic Science, FP = Food Production, PT = Pest Control, OCP = Other Cultural Practices; Gender (female = 0, male = 1); Diet (does not eat meat = 0, eats meat = 1); Pet (does not have a pet = 0, has a pet = 1).

\* $p < .05$

\*\* $p < .01$

\*\*\* $p < .001$.

of species ($F_{11, 3883} = 157.56$, MSe = 0.97, $p < 0.001$, $\eta_p^2 = .31$) and a main effect of purpose ($F_{3, 1115} = 69.66$, MSe = 4.87, $p < 0.001$, $\eta_p^2 = .16$). The means (s.e.m.) for each species were: carp = 2.58 (0.06); chicken = 2.66 (0.06); chimpanzee = 1.91 (0.05); dog = 1.98 (0.05); dolphin = 1.79 (0.05); frog = 2.46 (0.06); parrot = 1.97 (0.03); pig = 2.61 (0.06); pigeon = 2.63 (0.06); rabbit = 2.53 (0.06); rat = 2.81 (0.06); snake = 2.49 (0.06). Participants tended to disagree with the use of chimpanzees, dog, dolphins, parrot but be more neutral towards the use of carp, chicken, pig, pigeon, rabbit, frog, snake and rat. The means (s.e.m.) by purpose were: medical research = 2.76 (0.07); basic science = 2.68 (0.07); food production = 2.23 (0.05); pest control = 2.22 (0.05) and other cultural practices = 2.00 (0.06). Participants tended to be neutral to towards the use of animals for medical research and basic science but to disagree with animal use for food production, pest control and other cultural practices.

There were two significant two-way interactions: gender by animal ($F_{11, 3883} = 6.49$, MSe = 0.97, $p < 0.001$, $\eta_p^2 = .02$; see Fig 1) and animal by purpose ($F_{44, 15532} = 87.80$, MSe = 0.32, $p < 0.001$, $\eta_p^2 = .20$; Fig 2). For the gender by animal interaction, females were more likely to give pro-animal welfare scores compared to males for all animals, but this gender difference was larger for some species than others. The largest difference in ratings (.692) was for rabbits and the smallest was for dolphins (.233).

Exploring the animal by purpose interaction is more complicated, but some patterns do seem clear and consistent. The use of animals like pigs and chickens for food production was generally found to be more acceptable than all other animals where most people either

disagreed or strongly disagreed. An exception to this rule is carp which participants seemed more neutral about their use for the purpose of food production. For all animals except dolphins and chimpanzees participants tended to neither agree nor disagree with the use of animals for basic science or medical research. For dolphins and chimpanzees, participants tended to disagree to strongly disagree with their use for basic science or medical research. Generally, the use of animals in pest control was disagreed or strongly disagreed with for most types of animals, exceptions being rats, pigeons and snakes. For rats, pigeons and snakes, participants tended to express more neutral attitudes to their use in pest control. The use of animals in other cultural practices was generally disagreed with for all species.

The three way interaction of gender, animal and purpose was also significant ($F_{44, 15532} = 2.86$, MSe = 0.97 , $p < 0.001$, $\eta_p^2 = .008$; see Table 2). This indicates that there is a complex relation between animal and purpose which vary over males and females. However, as can be seen from Table 2, the overall effect is very small with a similar pattern across gender, therefore this interaction will not be discussed further.

Correlational and regression analyses to explore the interrelationships between questionnaire scores and demographic factors

To simplify the complexity of the results responses to the 12 animals used in the species questions were factor analysed for each of the five different purposes categories. Each purpose was dominated by a single factor (min % variance = 57%, max % variance = 79%). The average responses of the species for each of the five purposes was calculated and their reliabilities were assessed: medical research medical research α = .98, basic science α = .98, food production α = .93, pest control α = .94 and other cultural practices α = .97. Reliabilities were also calculated for each of the animal types across the five purposes. Internal consistency was good across all animal types with Cronbach's alphas in the range .81-.87. The overall reliability of the APQ scale was good, with a Cronbach's alpha of .98 suggesting high internal consistency between the items.

The correlations between the questionnaires (AAS and APQ) and relevant demographics are shown in Table 3. There was a positive association between diet and gender indicating that (in the larger online survey) males were more likely to eat meat however, there was no effect of gender on pet ownership, religion, ethnicity or age. The negative effect between gender and AAS scores and the positive effect between gender and the facets of the APQ suggests that females were more likely to show higher levels of animal welfare than males and be less willing to use animals regardless of the purpose. As shown in Table 3, diet had a negative effect on AAS scores and was positively correlated with all facets of the APQ questionnaire, indicating that those who eat meat expressed less animal welfare concerns and were more willing to endorse the use of animals. Those who had owned an animal were less likely to agree to use animals for medical research or basic science purposes than those who had never owned a pet. Those who were religious tended to have low scores on the AAS suggesting low animal welfare concerns and tended to give higher ratings for the use of animals for all purposes except food production. Participants reporting ethnicity other than White British showed higher levels of agreement with the use of animals for medical and basic science research, as well as for other cultural practices. Finally, older participants had lower scores on the AAS indicates less animal welfare concerns and higher scores on the use of animals for food production, pest control and other cultural practices. Education level received did not correlate with the AAS or APQ and was dropped from further analyses.

All the APQ scales were negatively correlated with AAS score (min $r = -.63$, max $r = -.75$). As expected more negative attitudes towards animals is associated with a greater willingness to use animals. All the APQ scales were positively associated with each other but varied in the

strength of their correlation (min $r$ = .55 max $r$ = .86) suggesting that the more inclined a person is to use an animal for one purpose the more inclined they are to use animals for other purposes too.

To explore how demographic factors influence the likelihood of using animals for medical, basic science, food production, pest control and other cultural practices five multiple linear regressions were performed with the five APQ purpose scales as the criterion variable. In each analysis, gender, diet, pet ownership, religion, ethnicity, and age were entered simultaneously in to the model (see Table 4).

All five regression models had significant F-tests indicating that they were a better fit ($p$ < .001) than the empty models. Each model accounted for 8–12% of the total variability in the APQ purpose subscales (medical research $R^2$ = .08, basic science $R^2$ = .12, food production $R^2$ = .11, pest control $R^2$ = .08, other cultural practices $R^2$ = .12). Gender was a significant positive predictor of all five APQ purpose of use scales indicating that males were more willing than females to use animals regardless of the purpose. Diet was also a significant positive predictor for all five APQ purposes scales suggesting that those who eat meat are more comfortable using animals regardless of purpose than those who do not eat meat. Owning a pet was a negative predictor of using animals for basic science meaning that those who own an animal were less willing for animals to be used for basic science. Ethnicity was a positive predictor for medical research, basic science and other cultural practices indicating that those of a non-white background were more willing to use animals for these purposes. Age was a positive predictor of basic science, pest control and other cultural practices suggesting that older participants were more comfortable using animals for these purposes than younger participants. Religion was not a significant predictor of animal use once other demographic variables were statistically controlled for.

## Discussion

The main aim of the present study was to develop and validate a questionnaire to measure attitudes to animal use in relation to species and the purpose of use. Although variation in such attitudes has previously been measured by other surveys and questionnaires, such as Ipsos MORI [8], the established AAS [4,29] and the PPP scale [32], none of these measures provides the same level of systematic comparison as the APQ. All participants additionally completed short versions of the AAS, to check concurrent validity.

Analyses confirmed that the APQ successfully differentiated attitudes to animal use across a variety of settings (e.g. medical research, food production) and types of animals (e.g., rodents, monkeys). The survey 1 pilot study results showed that there was overall less disagreement with the use of animals in medical research (mean rating 3.5) and basic science (mean rating 3.6), less endorsement for food production and pest control, and the use of animals for other cultural practices was generally disapproved irrespective of species. In survey 2, the levels of agreement were lower for both medical research (mean rating 2.76) and basic science (mean rating 2.68). However, as shown in Fig 2, the patterns of (dis-)agreement across purposes of use were largely replicated. The use of larger mammals (cats, dogs, horses and non-human primates) may be necessary before any human testing is permissible but amounts to a small proportion of overall usage of animals in biomedical research: 1% of all procedures in 2018. The use of dogs would seem likely to present a particular concern given their special status as 'man's best friend' but they remain in limited use (and with special protection) when no other species is suitable [12]. The results of the present study confirm that the use of dogs and non-human primates in biomedical research is indeed of particular public concern. Nonetheless, the findings contrast with the estimates of proportional concern derived from analyses of the

writings of animal rights activists, though this earlier study [13] did not distinguish medical versus non-medical research.

As might be expected, there was overall less support for the use of some non-domesticated and pet species. Attitudes to animal use showed the expected relationships with gender, femininity, diet and pet keeping. In the present study, there was no detectable effect of age or level of education. However, both of these variables were in a restricted range as the majority of the participants were university students. Similarly, we did not identify any effect of nationality, but the sample size was small in these demographic categories and we cannot exclude Type II error. The strong correlation between the levels of agreement with the use of animals as measured by the APQ and the AAS supports the conclusion that APQ has concurrent validity.

### Effects of purpose of use as measured by the APQ

Variation in attitudes towards purpose of animal use has been a particular focus in relation to medical research [8,31,43]. In the present study, medical research was also the purpose of use attracting least disagreement. However, it was clear from the open-ended comments that participants nonetheless had reservations about using animals in medical research; for example 'I agree with medical research using animals so long as it's necessary for the study, suffering is minimised, and if the consequence is preservation of human life. In other circumstances, I believe the use of animals is wrong.'

A commonly voiced opinion was that medical research may be acceptable when necessary and when the benefits outweigh the suffering of the animal. To the extent animal use for medical research successfully helps to improve and save humans lives, it presents a moral conflict for individuals who disagree with animal use in general. Agreement with the use of animals for basic science was non-significantly different from that expressed for the purpose of medical research. This lack of difference may reflect some appreciation of how medical advances are founded in basic science discovery and the importance of curiousity-driven research, which may also involve the use of invasive procedures and the killing of the animal upon the completion of laboratory-based studies. However, it could also reflect a lack of sufficient power, despite the larger sample used in survey 2, to identify a difference between basic and medical research (for which the effect size may be smaller).

With respect to food production, as might be expected, there were markedly different attitudes in relation to species. Participants rated the use of species commonly associated with food production (in the UK) as most acceptable, the use of monkeys, chimpanzees, dogs and dolphins was regarded as least acceptable. Considering the participant demographic was mostly British nationals, species categorised as pets would not be considered food, and this accounts for the high disapproval rating for the use of dogs (measured in survey 2). However, it is important to acknowledge that globally food preferences are not consistent and what might be considered culturally unethical in one community may be considered acceptable in another [24]. For example, in Asia, dog meat can be acceptable; in Africa, monkey is eaten as bush meat. However, these practices are these days very much in the minority, and usually only conducted by villagers who have a much lower level of education than an average university undergraduate. Consumption of wild animals may also be for medicinal purposes, including so-called black magic and shamanism. In addition, some religions require special treatment of particular animals: in Islam, pork is forbidden; in Hinduism, beef is prohibited due to the cow's sacred status [44]. To fully describe attitudes towards the use of animals in food production, would require more systematic comparison across different nationalities and cultures, and religious differences would need to be measured.

Pest control involves removing or killing animals which pose a risk to human health, the ecosystem or the economy. In the present study, agreement with the use of animals for the

purpose of pest control seemed to be influenced by the definition of what a pest is, which was in turn related to the species in question. For example, participants tended to disagree with killing as pests, animals which are typically viewed as domesticated in the UK. There was also evidence of how demographic factors may influence expressed attitudes in the open-ended comments: 'Living in a rural area has affected the way I see certain animals such as badgers, as I can see first-hand the consequence of the disease they spread. I also grew up round lots of hunting (fox, rabbit, pheasant); whilst I disagree with it, I still hold some sympathies.' (Survey 1).

Irrespective of species, 'other cultural practices' received the lowest overall approval ratings, in both surveys, compared to other purposes of use as defined for the APQ. This is consistent with earlier findings reported in the literature [31] as well as pressure for changes to the legislation to prohibit practices typically viewed as unnecessary and cruel uses of animals, for example bullfighting [45] and cosmetics testing [8]. The use of animals for the purpose of personal decoration similarly attracts high levels of public criticism, for example, to produce luxury clothing [3] or for other kinds of personal decoration [31]. As might be expected, there is also generally strong opposition to the use of animals for hunting, although some would see hunting as a form of pest control, to manage the number predators and thus protect other wildlife [46]. Respondents were asked to assume that all purposes involved harm, in order to match the outcomes for the animals across the purposes under examination. Cultural practices for which animals are used but not killed by (or upon completion of) the purpose of use (e.g., horse riding and wool production) were beyond the scope of the present study and we did not assess attitudes in relation to the pain levels that the animals were likely to experience [43].

## Species differences measured by the APQ

As might be expected, there was evidence for discrimination on the basis of species [16,17]. In particular, participants reported disagreement with the use of dogs (survey 2). Dogs are a prototypical pet species with which humans show close affectionate relationships [9,47]. Within the present sample, pet ownership had no significant effect on agreement with the use of animals for other purposes. However, the sample of participants without experience of pets was small: the majority of participants had experience of owning a pet and were British (Table 1). Britain is a pet-loving country with nearly a quarter of households owning a dog [48], and the overall profile of results is likely influenced by these demographics. The human-pet relationship likely varies with species. In Britain, dogs tend to attract the highest levels of regard as they are viewed as 'man's best friend'. Participants were also asked to rate their agreement with the use of rabbits (in both surveys 1 and 2). Although rabbits may be categorised as pets, they are also used for food consumption and in some cases may be regarded as pests. Much like the relationship with rabbits in Britain, dogs also can be viewed from very different perspectives in different cultures. For example, in China, dogs can be viewed as both pets and food [24].

The use of non-domesticated animals not categorised as pests might be expected to be rated similarly to the use of pet species, in that wildlife status might be expected to confer protection based on perception of importance and vulnerability [22]. Participants were disapproving of the killing of certain non-domesticated species, particularly in relation to food production, pest control and cultural practices, but were more neutral with respect to usage for medical research and basic science. Other animals which can be non-domesticated are perceived as pests, which may carry diseases which affect humans and thus have negative utilitarian value [32]. In the present study, participants were more inclined to endorse the use of rats and mice for purposes other than food production. It seems likely that rather than having moral objections to the use of 'pests' in food production, participants found the prospect of eating rodents

unappealing. Relatively higher levels of agreement with the use of pigs and chickens, species commonly associated with food production, also for other purposes, may be attributed to reduced ethical conflict for meat eating participants [27].

## Role of gender differences and levels of masculinity versus femininity

It has previously been reported that females report more pro-welfare animal attitudes than males [4,6,7]. This gender difference has also been examined in relation to the categorisation of animal species as pet, pest and profit [32,33], which relates to utility or purpose [22]. In the present study, females expressed less agreement than males with the use of mice, rats, rabbits, pigs, monkeys, octopus, chickens, badgers, zebrafish and tree shrews.

Analysis of the BSRI scores (survey 1 only) showed—perhaps paradoxically—that *both* masculinity and femininity were associated with pro-welfare attitudes as measure by the AAS. BSRI femininity predicted disagreement with animal use as measured by the APQ. Perhaps surprisingly, neither the masculinity nor the femininity BRSI scores showed a significant relationship with any of the APQ purpose of use scales when gender was statistically controlled. This suggests that gender is a better predictor of attitudes to animal use than is a more differentiated measure of gendered behaviours and personality traits. In the present study, the majority of participants were female, and would therefore be expected to be in overall less agreement with the use of animals. However, the fact that the samples were both predominantly female cannot account for the differing profiles of responses by species and purpose of use (shown in Fig 1 and Table 2).

## Other demographic characteristics

It has previously been reported that younger individuals generally show greater pro-welfare attitudes compared to older people [7,8,22]. However, the effect of age has been found to depend on the motivation for the attitude [21] and increasing age predicts increased agreement with the use of animals for research purposes [8]. The present samples were relatively young (mostly 21 and under in survey 1 and on average 34 years old in survey 2) but nonetheless relatively pro-research as compared with the other suggested purposes.

In addition, individuals growing up in different countries and cultures will have widely different experiences in relation to the use of animals [49,50]. Western nationalities tend to view animal suffering more negatively, particularly for pet species, showing higher disagreement with the use of animal products for personal decoration, yet (at the same time) relatively high levels of support for the use of animals in scientific research. In the present study, we found no effect of nationality. However, (as above) such negative findings can be attributed to lack of variation in the sample and/or Type II error.

As might be expected, previous studies have found that diet predicts attitudes to animal use, especially in the case of vegetarians and vegans [27, 51], particularly in relation to levels of agreement with eating animals. Studies have also shown that compared to meat-eaters, vegetarians also display higher levels of concern for animals used in research [5], particularly those vegetarians motivated primarily by ethical rather than health concerns [27]. In the present study, diet was generally predictive of reduced levels of agreement with the use of animals, and (with the exception of other cultural practices in survey 1) across all the suggested purposes. Likely differences by species are beyond the scope of the present study. For example, attitudes towards animals viewed as pets tend not to be influenced by dietary factors, presumably because the relationship with pet species is based on companionship which is incompatible with other kinds of utility. Raising an animal for food production is contextually very different to owning an animal as a pet which becomes part of the household [9,52]. In 2016, the pet

population in the UK totalled 57 million [48]. Dogs are one of the most common companion animals associated with humans, and the history of this companionship dates back over 12,000 years [47]. Individuals who experience having a pet tend to hold more positive attitudes towards both pet and non-pet species [53], including 'unpopular' species typically perceived as predators, pests or disease-carrying [52]. In survey 1, pet keeping was generally predictive of reduced levels of agreement with the use of animals for medical research or basic science purposes. Interestingly, and possibly related to the use of a more diverse sample, the effect of pet keeping in survey 2 was even more restricted (to the purpose of basic science).

## Limitations of the present study

The APQ is a new scale and responses in relation to purpose of use may have been influenced by the specific examples given. In particular, examples of other cultural practices may have resulted in a particularly negative view of the use of animals for that category of purpose, intended to cover hunting, animal fighting and other forms of animal entertainment, such as zoos and circuses as well as wearing (the skin) as clothing or as ornamentation. Moreover, positive uses of animals such as pet-keeping and bird feeding should be seen as other cultural practices, but no such positive examples were provided to illustrate this category.

The final selections of species used in surveys 1 and 2 were arbitrary, not fully representative and certainly not an exhaustive list. In the same way we made some changes to the species selected for surveys 1 and 2 we hope that others will adopt and adapt the proposed questionnaire format to suit their research purposes. For example, invertebrates have been understudied to date and this omission should be rectified in future studies.

Going forward the scale will require further validation, e.g. test-retest reliability, and in additional non-student samples [11,54]. It is an omission that we were unaware of the new Speciesism Scale [55] at the time the study was run, concurrent validity with this scale should also be determined in future studies. However, the APQ does show concurrent validity with the established AAS [29]. Moreover, unlike alternative scales which use conventionally formulated items [4,29,55], the APQ has a number of advantages in that it is designed for systematic comparison across species and purpose of use. Moreover, as the *profile* of agreement/disagreement, rather than the absolute levels of (dis-)agreement with the use of different species is the focus of interest, social desirability biases and differences due to population baseline are controlled for.

In line with majority of the previous research on attitudes to animal use, in which psychology students are commonly sampled [7], the demographic profile of the typical survey 1 participant was a female psychology student, of in this case British nationality, aged 21 or under, ate meat and had experience of owning a pet. Thus, the findings of survey 1 may lack generality to wider populations since the sample was primarily composed of young, female, western and educated individuals [54]. Survey 2 which was conducted online reached a wider demographic: participants had a mean age of over 33 years and were 29% non-British. However, the sample was still primarily composed of British participants and the findings cannot be assumed to generalise to other nationalities. Moreover, the number of bivariate correlations conducted increases the risk that some effects may be spurious. However, the predicted effects of gender and diet on the APQ and AAS scores were seen consistently across the two surveys. This consistency of outcomes—despite the differences in survey methodology–suggests that these findings are robust.

Finally, the focus of the APQ was on species and purpose of use per se rather involving and judgement of the levels of suffering experienced, as addressed by the refinement of experimental procedures [56]. Possibly to focus on levels of suffering, would have resulted in a different profile of levels of (dis-)agreement [17].

## Conclusions

Notwithstanding its limitations, the present study provides some benchmark data on attitudes to animals by species and purpose of use. The new APQ scale is readily adapted to obtain comparative data on other species, which could be useful in the context of replacement [56], when animals—which would be interchangeably suitable from a scientific perspective—might be expected to arouse different levels of public sympathy. For example, the use of pigs in place of dogs has been found to attract reduced criticism [23]. The APQ can be used to provide further systematic evidence on which animal species are considered suitable for which purposes of use and by which publics. With respect to the demographic factors, the strongest predictor was gender. In general, females express less agreement with the use of animals. After controlling for this effect of gender, diet continued to be predictive in both surveys, and showed some discrimination by purpose of use. Specifically, diet is predictive for all purposes of use except for other cultural practices (in survey 1); meat eaters tend to be relatively more accepting of the use of animals for basic science, medical research, food production and pest control. The effects of pet ownership seen in survey 1 were partially reproduced in survey 2, for basic science but not medical research.

Thus the APQ provides a new tool to unpack how public attitudes depend on the intersectionality of demographics, species and purpose of use. Measures of intersectionality are needed so that we can better quantify and contextualise the wide variability in public concern for animals used for different purposes, particularly given that levels of concern can be quite disproportionate in relation to the numbers of animals involved [13]. The APQ also provides a potentially useful tool to measure changes in reported attitudes, specifically in relation to species and purpose, following interventions such outreach activities intended to prompt reflection on the use of animals for research purposes, for example. Importantly, changes in the profile of attitudes to animal use by species and purpose control for any general demand characteristic of the repeated testing which might shift ratings in an overall pro-welfare direction.

## Supporting information

**S1 Appendix. The presentation format used for the animal purpose questionnaire (APQ) in survey 1.** The question 'How do you feel about the use of. . .' was repeated in the identical format for 10 types of animal. Thus participants were asked to repeat their ratings also for **rats**, **rabbits**, **pigs**, **monkeys**, **octopus**, **chickens**, **badgers**, **zebrafish** and **tree shrews** (just the animal to be considered was changed in the question line).
(DOCX)

**S2 Appendix. The Qualtrics presentation format for the animal purpose questionnaire (APQ) in survey 2.** The question 'To what extent do you agree with the use of. . .' was repeated in the identical format for the 5 categories of purpose. Thus participants were asked to repeat the ratings also for **BASIC SCIENCE RESEARCH**, **FOOD PRODUCTION**, **PEST CONTROL** and **OTHER PRACTICES**.
(DOCX)

**S1 Data. Excel file Survey1Data contains the raw data for survey 1 labelled item by item with the question number, the first few letters of the item or species.** Items beginning with BS are from the Bem Sex-Role Inventory and items beginning with H are from Herzog et al.'s Animal Attitudes Scale [29]. Items from the Animal Purpose Questionnaire are labelled by species and purpose. Other abbreviations are as in the text or should be self-explanatory.
(CSV)

**S2 Data. Excel file Survey2Data contains the Qualtrics output from Survey 2, anonymised by removal of the participants' location tags and the nicknames they provided.** Row 1 provides the question number and some descriptor. Items beginning with H relate Herzog et al.'s Animal Attitudes Scale [29]. Items labelled with AES are from the Animal Empathy Scale (data not reported). Items beginning with APQ originate from the Animal Purpose Questionnaire and BAM refers to items from the Belief in Animal Mind Scale.
(CSV)

## Acknowledgments

We thank Grace Caton, Rachel Smith, Bethany Brown and Charlotte Wastell for assistance with data collection.

## Author Contributions

**Conceptualization:** Peter A. Bibby, Helen J. Cassaday.

**Data curation:** Alexander Bradley, Peter A. Bibby, Helen J. Cassaday.

**Formal analysis:** Alexander Bradley, Peter A. Bibby.

**Investigation:** Neil Mennie, Helen J. Cassaday.

**Methodology:** Neil Mennie, Helen J. Cassaday.

**Project administration:** Neil Mennie, Helen J. Cassaday.

**Resources:** Helen J. Cassaday.

**Supervision:** Helen J. Cassaday.

**Writing – original draft:** Alexander Bradley, Peter A. Bibby, Helen J. Cassaday.

**Writing – review & editing:** Alexander Bradley, Neil Mennie, Helen J. Cassaday.

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
