## [Decision Letter · Decision Letter 0]

15 Oct 2019

PONE-D-19-23588

Some animals are more equal than others: Validation of a new scale to measure how attitudes to animals depend on species and human purpose of use.

PLOS ONE

Dear Dr Cassaday,

Thank you for submitting your manuscript to PLOS ONE. After careful consideration, we feel that it has merit but does not fully meet PLOS ONE’s publication criteria as it currently stands. Therefore, we invite you to submit a revised version of the manuscript that addresses the points raised during the review process.

Please consider the reviewer comments below, and explain in your response how they have been taken into account in the revision.

We would appreciate receiving your revised manuscript by Nov 29 2019 11:59PM. To enhance the reproducibility of your results, we recommend that if applicable you deposit your laboratory protocols in protocols.io, where a protocol can be assigned its own identifier (DOI) such that it can be cited independently in the future. For instructions see: http://journals.plos.org/plosone/s/submission-guidelines#loc-laboratory-protocols

We look forward to receiving your revised manuscript.

Kind regards,

I Anna S Olsson, Ph.D.

Academic Editor

PLOS ONE

Journal Requirements:

Reviewers' comments:

Reviewer's Responses to Questions

**Comments to the Author**

1. Is the manuscript technically sound, and do the data support the conclusions?

Reviewer #1: Yes

Reviewer #2: Yes

2. Has the statistical analysis been performed appropriately and rigorously? 

Reviewer #1: No

Reviewer #2: Yes

3. Have the authors made all data underlying the findings in their manuscript fully available?

Reviewer #1: Yes

Reviewer #2: Yes

4. Is the manuscript presented in an intelligible fashion and written in standard English?

Reviewer #1: Yes

Reviewer #2: Yes

5. Review Comments to the Author

Reviewer #1: This is the second time I have reviewed this manuscript for PLOS ONE. This version is much improved, and the reviewers addressed the issues that I had with the first version.

Statistical analysis: I am concerned the lack of information on the reliability of the APQ. It is standard practice to include information on the internal reliability of new scales using, for example, Cronbach’s alpha. The authors mention that the AAS=10 has high reliability (Cronbach’s alpha), but this is based on single scores calculated by summing responses to the 10 AAS items. As far as I could tell, no information is given on the reliabilities of the various subscales of the APQ. This information could be included in a separate table.

While it is too late now, several of the descriptions of the use categories could be improved. For instance, the term “Other Cultural Practices” covers a lot of territory. The exemplar in the scale is “using body parts as ornaments.” And the text of the paper mentions bullfighting as an example. However, positive uses of animals such as pet-keeping and bird feeding are also “other cultural practices.” Similarly, I am not sure that “bushmeat” is the best example of the use of animals for food.

Comparisons between the AAS and the APQ are frequently made in the manuscript. The problem is that the AAS is scored so that individuals who are concerned with animal welfare (that is, generally oppose the use of animals) get high scores. However, the APQ is scored so that individual who generally disapprove various uses of animals and species get low scores. Hence, AAS and APQ scores are highly related – but in in the opposite directions.

This can create confusion even in the present ms. For example, Lines 467 – 469 state “Diet had a negative effect on AAS scores and all facets of the APQ questionnaire indicating that those who eat meat have less animal welfare concerns and were more willing to use animals.” Yet according to Table 3b, the APQ scores were actually correlated positively with meat-eating – as would be expected.)

For this reason, it might be a good idea to include a paragraph emphasizing that the differences in the scoring of these scales. Even thought the scales assess similar underlying dimensions of attitudes toward animals, the APQ is a measure of the degree of approval of various uses of animal by humans while the AAS measures concern for animal used in various situations.

Minor suggestions

-When citing papers in Anthrozoos, there is no need to include “A Multidisciplinary Journal of the Interactions of People and Animals” in the citation.

-The new Speciesism Scale by Caviola et al. covers much of the same territory of the AAS and the APQ. It should be cited and possibly discussed. The reference is Caviola et al (2019) The moral standing of animals: Towards a psychology of speciesism. J Pers Soc Psychol. 116(6), 1011-1029.

Table 2 - Indicate what the subscale initials mean on the captions (MR, BS, FP etc.) This is done in the other tables.

Animals. I was surprised the lack of variation in some of the species. It might be interesting in future administrations to include, say, fruit flies.

Reviewer #2: Overall impression - The submitted manuscript describes the development of a new attitude to animal scale in relation to species and purpose of use, the Animal Purpose Questionnaire. Building upon a previous submission, the authors have submitted the APQ to two different populations, together with the established Attitudes to Animals Scale. Despite these efforts, some methodological concerns remain.

The introduction is both informative and concise.

The authors have revised the manuscript by providing further data. There are now results from two different sets of studies: one smaller pen and paper survey to a population mostly made of female, 21 or under, psychology British students (N=128), and a wider on-line survey to 419 participants (most participants were female, most own pets and there were 39 nationalities represented). Both populations were conveniently sampled. Considering that the attitudes to animals are often culturally constructed, the high number of different nationalities may pose a challenge for the interpretation and generalization of results. The low proportion of men and respondents not owning pets are also limiting factors (the authors acknowledge these limitations).

Although both studies yield comparable results, I am not sure that presenting them together is the best approach. In fact, methodological differences between both surveys render meaningful comparisons problematic. First there is the imprinting effect from the examples. The examples of purpose of use in survey 1 (e.g. bush meat for food production) differed from those in survey 2. In addition, the difference between medical research and basic science may not be entirely clear for most participants (even I was unsure of their remit). Moreover, the changes made to the wordings and scoring in survey 2 must be considered. For example, removing the ‘Not applicable’ option in survey 2 is not without its flaws. Finally, there is the issue of survey mode of administration: pen & paper surveys are known to elicit different response biases than online surveys. I would suggest considering survey 1 as a pilot/development study (and publishing it as such), and survey 2 as the validation study, which has been improved in accordance to the results of survey 1.

In terms of results, the large number of variables being calculated raises some concerns regarding p-hacking. For example, for the comparison between AAS/APQ scores and demographic variables, generated 110 correlations alone (Table 3), but not all are equally relevant, and some may even be spurious (e.g. with such a predominantly British population, there is little use of analysing the effect of nationality). This generates some noise in the discussion when trying to explain differences between survey 1 and the more methodologically robust survey 2. Finally, the fact that the use of animals for basic science did not differ significantly from their use for medical research may be no more than a reflection of the lack of power of the APQ in differentiating between the two.

6. PLOS authors have the option to publish the peer review history of their article (what does this mean?). If published, this will include your full peer review and any attached files.

Reviewer #1: No

Reviewer #2: Yes: Manuel Magalhaes Sant'Ana

---

## [Author Response · Author response to Decision Letter 0]

31 Oct 2019

Ref: PONE-D-19-23588

Title: Some animals are more equal than others: Validation of a new scale to measure how attitudes to animals depend on species and human purpose of use.

Journal: PLOS ONE

Journal Requirements

We have checked the PLOS ONE style requirements and labelled the files as per the templates.

Captions for and in text citations of the Supporting Information files have been provided.

Response to Reviewers 

The checkbox responses are positive with the exception of Q2 (Reviewer #1). The specific points concerning the statistical analysis will be addressed below.

Reviewer #1: 

This is the second time I have reviewed this manuscript for PLOS ONE. This version is much improved, and the reviewers addressed the issues that I had with the first version.

We are pleased that the reviewer finds the manuscript much improved and finds the initial issues raised to have been addressed.

Statistical analysis: I am concerned the lack of information on the reliability of the APQ. It is standard practice to include information on the internal reliability of new scales using, for example, Cronbach’s alpha. The authors mention that the AAS=10 has high reliability (Cronbach’s alpha), but this is based on single scores calculated by summing responses to the 10 AAS items. As far as I could tell, no information is given on the reliabilities of the various subscales of the APQ. This information could be included in a separate table.

We appreciate that information on reliability might normally be expected to appear in the Methods. However, since it needed to follow on from the factor analysis, the Cronbach’s alphas for the five purpose subscales of the APQ were reported in the Results part of the manuscript (p15 under the subsection of ‘Correlation and regression analyses to explore the interrelationship between questionnaire scores and demographic factors). In addition, we have now added a sentence highlighting the overall reliability of the scales for studies one and two (both � = .98; Study 2: � = .98). We have also added a sentence explaining the reliabilities by animals across purposes for surveys one and two (p20). 

While it is too late now, several of the descriptions of the use categories could be improved. For instance, the term “Other Cultural Practices” covers a lot of territory. The exemplar in the scale is “using body parts as ornaments.” And the text of the paper mentions bullfighting as an example. However, positive uses of animals such as pet-keeping and bird feeding are also “other cultural practices.” Similarly, I am not sure that “bushmeat” is the best example of the use of animals for food.

The limitation that the examples provided to illustrate the category of other cultural practices were all negative has been acknowledged p29. As explained p11, the wording of the examples given to illustrate the purposes of use was improved in survey 2 (the reference to bush meat was dropped).

Comparisons between the AAS and the APQ are frequently made in the manuscript. The problem is that the AAS is scored so that individuals who are concerned with animal welfare (that is, generally oppose the use of animals) get high scores. However, the APQ is scored so that individual who generally disapprove various uses of animals and species get low scores. Hence, AAS and APQ scores are highly related – but in in the opposite directions.

The expected direction of association between the scales has been clarified p12.

This can create confusion even in the present ms. For example, Lines 467 – 469 state “Diet had a negative effect on AAS scores and all facets of the APQ questionnaire indicating that those who eat meat have less animal welfare concerns and were more willing to use animals.” Yet according to Table 3b, the APQ scores were actually correlated positively with meat-eating – as would be expected.)

The statement of this result has been clarified (with reference to Table 3) p21.

For this reason, it might be a good idea to include a paragraph emphasizing that the differences in the scoring of these scales. Even though the scales assess similar underlying dimensions of attitudes toward animals, the APQ is a measure of the degree of approval of various uses of animal by humans while the AAS measures concern for animal used in various situations.

Apologies for the error, we have now corrected the effects of diet on APQ to be in agreement with Table 3 (p21). We agree with the reviewer’s point that adding a paragraph emphasizing the different scoring of the scales would help with the clarity of the manuscript. This has been done p12.

Minor suggestions

-When citing papers in Anthrozoos, there is no need to include “A Multidisciplinary Journal of the Interactions of People and Animals” in the citation.

The citations to Anthrozoos have been corrected throughout the reference list. Other references have also been reformatted to match journal style.

-The new Speciesism Scale by Caviola et al. covers much of the same territory of the AAS and the APQ. It should be cited and possibly discussed. The reference is Caviola et al (2019) The moral standing of animals: Towards a psychology of speciesism. J Pers Soc Psychol. 116(6), 1011-1029.

The suggested reference has been cited. The APQ is much simpler in conception, in that we examine attitudes to animal use systematically by species and purpose of use. Since it is any difference in ratings by species and/or purpose of use which is of interest we have an inbuilt control for the demand characteristics of the scale. The Speciesism Scale uses more conventionally formulated items (some modified from existing scales), followed by factor analysis. In contrast, the APQ has the a priori factors of species and purpose. We agree that the Speciesism Scale presents a very valuable addition to the literature and this has been highlighted p30 but the approach taken in the present study is quite different.

Table 2 - Indicate what the subscale initials mean on the captions (MR, BS, FP etc.) This is done in the other tables.

The Table 2 abbreviations are highlighted in the revision (they might slip off the page if the format changes in file conversion).

Animals. I was surprised the lack of variation in some of the species. It might be interesting in future administrations to include, say, fruit flies.

We agree that the selections of species were arbitrary and certainly not an exhaustive list. In the same way we made some changes to the species selected for surveys 1 and 2 we hope that others will adopt and adapt the proposed questionnaire format to suit their research purposes. We agree that invertebrates have been understudied to date and that this omission should be rectified in future studies. This limitation has been acknowledged p30.

Reviewer #2: 

Overall impression - The submitted manuscript describes the development of a new attitude to animal scale in relation to species and purpose of use, the Animal Purpose Questionnaire. Building upon a previous submission, the authors have submitted the APQ to two different populations, together with the established Attitudes to Animals Scale. Despite these efforts, some methodological concerns remain. The introduction is both informative and concise.

We are pleased that the reviewer finds some improvement in the manuscript, e.g. in the introduction.

The authors have revised the manuscript by providing further data. There are now results from two different sets of studies: one smaller pen and paper survey to a population mostly made of female, 21 or under, psychology British students (N=128), and a wider on-line survey to 419 participants (most participants were female, most own pets and there were 39 nationalities represented). Both populations were conveniently sampled. Considering that the attitudes to animals are often culturally constructed, the high number of different nationalities may pose a challenge for the interpretation and generalization of results. The low proportion of men and respondents not owning pets are also limiting factors (the authors acknowledge these limitations).

As discussed p30, we agree that there is further work to be done to test the generality of the findings.

Although both studies yield comparable results, I am not sure that presenting them together is the best approach. In fact, methodological differences between both surveys render meaningful comparisons problematic. First there is the imprinting effect from the examples. The examples of purpose of use in survey 1 (e.g. bush meat for food production) differed from those in survey 2. In addition, the difference between medical research and basic science may not be entirely clear for most participants (even I was unsure of their remit). Moreover, the changes made to the wordings and scoring in survey 2 must be considered. For example, removing the ‘Not applicable’ option in survey 2 is not without its flaws. Finally, there is the issue of survey mode of administration: pen & paper surveys are known to elicit different response biases than online surveys. I would suggest considering survey 1 as a pilot/development study (and publishing it as such), and survey 2 as the validation study, which has been improved in accordance to the results of survey 1.

Our justifications to include survey 1 include the demonstrated reproducibility of some of the main findings shown in survey 2 despite the methodological differences (beyond presentation format) identified by the reviewer. The survey 2 data show broadly the same profile of responses across the purposes of use. The effects of sex and diet on the APQ and AAS scores are also consistent across the two surveys. The point of the comparison notwithstanding, these differences have been further acknowledged p30. Survey 1 also shows the viability of administering the questionnaire in paper format, paper format can be preferable for example to assess the effectiveness of outreach activities.

We take the point that survey 1 can be referred to as a pilot/development study and survey 2 as the validation study and this terminology has been adopted in the revised manuscript (p7 and elsewhere).

In terms of results, the large number of variables being calculated raises some concerns regarding p-hacking. For example, for the comparison between AAS/APQ scores and demographic variables, generated 110 correlations alone (Table 3), but not all are equally relevant, and some may even be spurious (e.g. with such a predominantly British population, there is little use of analysing the effect of nationality). This generates some noise in the discussion when trying to explain differences between survey 1 and the more methodologically robust survey 2. Finally, the fact that the use of animals for basic science did not differ significantly from their use for medical research may be no more than a reflection of the lack of power of the APQ in differentiating between the two.

The point about noise in the discussion has in part been addressed by the revised terminologies for the surveys. The lack of difference seen in attitudes to using animals for basic science vs medical research could reflect lack of power but the statistical power was quite sufficient to detect differences in attitudes in relation to other purposes of use. Even if it’s just a smaller effect size for the difference between medical research and basic science we’d still see that as of interest. Still we have added the suggested caveat p24.

We agree that a large number of bivariate correlations raises the risk that some effects may be spurious. We wish to point to a number of key findings which replicated across the surveys, please see also above response to the point made about comparing across the surveys.

However, we agree that a note of caution is warranted in the interpretation of the findings so we raise the reviewer’s point in the limitations section (p30).

---

## [Decision Letter · Decision Letter 1]

2 Dec 2019

PONE-D-19-23588R1

Some animals are more equal than others: Validation of a new scale to measure how attitudes to animals depend on species and human purpose of use.

PLOS ONE

Dear Dr Cassaday,

Thank you for submitting your manuscript to PLOS ONE. After careful consideration, we feel that it has merit but does not fully meet PLOS ONE’s publication criteria as it currently stands. Therefore, we invite you to submit a revised version of the manuscript that addresses the points raised during the review process.

Your revision meets the reviewer comments from the previous rounds, and my thorough reading of the revised version picked up a couple of minor inconsistencies and/or typos:

General: There seem to be an underlying assumption that is never made explicit: that animals are always killed or in other ways significantly harmed for the purposes you are including in your study. From the way you discuss the results, especially on lines 565-574 and 592-611, I assume this is how you have thought about the purposes. There are cultural practices for which animals are used but not necessarily significantly harmed (e g sport horses) and it is also possible to get clothing from animals without harming them (wool). Please add a brief reflection on this aspect, and whether or not it was made explicit to the respondents.

Line 63 Not all readers are familiar with CITES, please add a brief explanation 

Line 11 A word seems to be missing here

Line 118 Non-British readers will not be familiar with the Ipsos MORI polls, please add a brief explanation

Lines 201-202 A word or part of a sentence seems to be missing here: the sentence ends with a right parenthesis but there is no left parenthesis.

Lines 257-258 Please add a brief explanation of what the BSRI scale measures. Is "masculine traits" to be understood as "traits traditionally associated with masculinity"? Also, are these physical or psychological attributes?

Throughout the Results section, you refer almost consistently to "sex" and not "gender". In the Methods section, you refer to "gender", which seems to me to be the correct term for what you deal with here, presumably self-identified. Also, do you only have "male" and "female"?

Line 358 word missing

Line 411 "don't eat meat" should be "do not eat meat"

Line 559 You may want to refer to the work by Thomas Boker Lund on the relation between different purposes and acceptance of animal research DOI: 10.1177/0963662512451402

Line 663 With the measure taken out of context as here it is not obvious how "on average over 33 years old" is an indicator of that the sample is young, as the only thing this number tells us is that on average the sample is older rather than younger!

Line 673 Please also refer to the work by Thomas Boker Lund on the animal ethics profile of vegetarians, vegans and meat-eaters DOI: 10.1080/08927936.2015.1083192

Line 711-712 word missing

Please remove the word count from the abstract and the full text.

We would appreciate receiving your revised manuscript by Jan 16 2020 11:59PM. To enhance the reproducibility of your results, we recommend that if applicable you deposit your laboratory protocols in protocols.io, where a protocol can be assigned its own identifier (DOI) such that it can be cited independently in the future. For instructions see: http://journals.plos.org/plosone/s/submission-guidelines#loc-laboratory-protocols

We look forward to receiving your revised manuscript.

Kind regards,

I Anna S Olsson, Ph.D.

Academic Editor

PLOS ONE

Reviewers' comments:

Reviewer's Responses to Questions

**Comments to the Author**

1. If the authors have adequately addressed your comments raised in a previous round of review and you feel that this manuscript is now acceptable for publication, you may indicate that here to bypass the “Comments to the Author” section, enter your conflict of interest statement in the “Confidential to Editor” section, and submit your "Accept" recommendation.

Reviewer #2: All comments have been addressed

2. Is the manuscript technically sound, and do the data support the conclusions?

Reviewer #2: Yes

3. Has the statistical analysis been performed appropriately and rigorously? 

Reviewer #2: Yes

4. Have the authors made all data underlying the findings in their manuscript fully available?

Reviewer #2: Yes

5. Is the manuscript presented in an intelligible fashion and written in standard English?

Reviewer #2: Yes

6. Review Comments to the Author

Reviewer #2: I have now read through the authors’ response to the reviews and the revised submission. I consider that the revised manuscript can be accepted.

7. PLOS authors have the option to publish the peer review history of their article (what does this mean?). If published, this will include your full peer review and any attached files.

Reviewer #2: Yes: Manuel Magalhaes-Sant'Ana

---

## [Author Response · Author response to Decision Letter 1]

2 Jan 2020

PONE-D-19-23588R1

Some animals are more equal than others: Validation of a new scale to measure how attitudes to animals depend on species and human purpose of use.

Response document

Reviewer #2 is fully satisfied and considers that the revised manuscript can be accepted. There are no further comments from Reviewer #1.

The further revised version of the manuscript addresses the minor points for clarification, inconsistencies and/or typos identified by the Academic Editor. We have also added the suggested references.

General: There seem to be an underlying assumption that is never made explicit: that animals are always killed or in other ways significantly harmed for the purposes you are including in your study. From the way you discuss the results, especially on lines 565-574 and 592-611, I assume this is how you have thought about the purposes. There are cultural practices for which animals are used but not necessarily significantly harmed (e g sport horses) and it is also possible to get clothing from animals without harming them (wool). Please add a brief reflection on this aspect, and whether or not it was made explicit to the respondents.

This has been clarified in the Methods p11. Respondents were asked to assume that all purposes involved harm, in order to match the outcomes for the animals across the purposes under examination. Specifically both APQ surveys were prefaced with the instruction ‘…rate whether you agree or disagree with the killing of different types of animal for the following purposes:…’

In the revised discussion p26 we now flag that cultural practices for which animals are used but not necessarily significantly harmed (e.g., horse riding and wool production) were beyond the scope of the present study. The discussion has also been clarified p25.

Line 63 Not all readers are familiar with CITES, please add a brief explanation

This has been done p3. CITES (the Convention on International Trade in Endangered Species of Wild Fauna and Flora) has published a list of specially protected species.

Line 11 A word seems to be missing here

This has been corrected by insertion of ‘of’ (their use) p5 (line 111)

Line 118 Non-British readers will not be familiar with the Ipsos MORI polls, please add a brief explanation

This has been done p5. Ipsos MORI is a UK-based market research company which conducts independent surveys for a wide range of organisations.

Lines 201-202 A word or part of a sentence seems to be missing here: the sentence ends with a right parenthesis but there is no left parenthesis.

This has been corrected p9.

Lines 257-258 Please add a brief explanation of what the BSRI scale measures. Is "masculine traits" to be understood as "traits traditionally associated with masculinity"? Also, are these physical or psychological attributes?

This has been done p11. The BSRI provides a measure of psychological profile and we have rephrased the description of masculine and feminine traits as suggested.

Throughout the Results section, you refer almost consistently to "sex" and not "gender". In the Methods section, you refer to "gender", which seems to me to be the correct term for what you deal with here, presumably self-identified. Also, do you only have "male" and "female"?

In the revised manuscript, the term gender has now been used consistently throughout.

In both surveys, participants were presented with four options in relation to gender: male, female, ‘other’ and ‘prefer not to say’. In survey 1, all participants self-identified themselves as male or female. In survey 2, one selected ‘other’ and two participants selected ‘prefer not to say.’ This information has been provided in the revision p9 but we cannot classify these participants as non-binary. As only very small numbers of participants identified as ‘other’ or ‘prefer not to say’, these participants were excluded for analyses by gender.

Line 358 word missing

This has been corrected by insertion of ‘of’ (the animal types) p15

Line 411 "don't eat meat" should be "do not eat meat"

This has been corrected p18.

Line 559 You may want to refer to the work by Thomas Boker Lund on the relation between different purposes and acceptance of animal research DOI: 10.1177/0963662512451402

This additional study has been cited as appropriate p24 (in the context of attitudes to animal use in relation to medical research). We don’t further elaborate here as the purposes examined were all within the general category of medical research and testing (cancer, cardio-vascular, migraine, obesity, cosmetic testing). However, we do further cite this study p26 as (in contrast to the approach taken in the present study) attitudes were assessed in relation to the pain levels that the animals were likely to experience.

[43] Lund, T.B., Mørkbak, R.M., Lassen, J., & Sandøe, P. Painful dilemmas: A study of the way the public’s assessment of animal research balances costs to animals against human benefits. Public Underst Sci. 2014;23(4): 428-444. doi: 10.1177/0963662512451402

Line 663 With the measure taken out of context as here it is not obvious how "on average over 33 years old" is an indicator of that the sample is young, as the only thing this number tells us is that on average the sample is older rather than younger!

Apologies, the intended point has been clarified p28. The average was just over 33 years for Survey 2 (33.84 as per Table 1, rounded to ‘on average 34 years old’ for the purposes of the discussion in the revision).

Line 673 Please also refer to the work by Thomas Boker Lund on the animal ethics profile of vegetarians, vegans and meat-eaters DOI: 10.1080/08927936.2015.1083192

[51] Lund, T.B., McKeegan, D.E.F., Cribbin, C., & Sandøe, P. Animal ethics profiling of vegetarians, vegans and meat-eaters. Anthrozoos. 2016;29(1): 89-106. doi: 10.1080/08927936.2015.1083192

 This additional study has been cited p29.

Line 711-712 word missing

This has been corrected by insertion of ‘rather’ (than) and some further highlighted clarification p30.

Please remove the word count from the abstract and the full text.

The word counts have been removed.

To enhance the reproducibility of your results, we recommend that if applicable you deposit your laboratory protocols in protocols.io, where a protocol can be assigned its own identifier (DOI) such that it can be cited independently in the future. For instructions see: http://journals.plos.org/plosone/s/submission-guidelines#loc-laboratory-protocols

Not applicable: there was no laboratory protocol for a study of this kind.

---

## [Editor Report · Decision Letter 2]

6 Jan 2020

Some animals are more equal than others: Validation of a new scale to measure how attitudes to animals depend on species and human purpose of use.

PONE-D-19-23588R2

Dear Dr. Cassaday,

We are pleased to inform you that your manuscript has been judged scientifically suitable for publication and will be formally accepted for publication once it complies with all outstanding technical requirements.

With kind regards,

I Anna S Olsson, Ph.D.

Academic Editor

PLOS ONE
---

## [Editor Report · Acceptance letter]

10 Jan 2020

PONE-D-19-23588R2 

Some animals are more equal than others: Validation of a new scale to measure how attitudes to animals depend on species and human purpose of use. 

Dear Dr. Cassaday:

I am pleased to inform you that your manuscript has been deemed suitable for publication in PLOS ONE. Congratulations! Your manuscript is now with our production department. 

With kind regards,

on behalf of

Dr I Anna S Olsson 

Academic Editor

PLOS ONE